# Pterostilbene, a Dimethyl Derivative of Resveratrol, Exerts Cytotoxic Effects on Melanin-Producing Cells through Metabolic Activation by Tyrosinase

**DOI:** 10.3390/ijms25189990

**Published:** 2024-09-17

**Authors:** Hitomi Tanaka, Tomoko Nishimaki-Mogami, Norimasa Tamehiro, Norihito Shibata, Hiroki Mandai, Shosuke Ito, Kazumasa Wakamatsu

**Affiliations:** 1Department of Medical Technology, School of Health Sciences, Gifu University of Medical Science, 795-1 Nagamine, Ichihiraga, Seki 501-3892, Japan; hitanaka@u-gifu-ms.ac.jp; 2Institute for Melanin Chemistry, Fujita Health University, 1-98 Dengakugakubo, Kutsukake-cho, Toyoake 470-1192, Japan; 3Division of Biochemistry, National Institute of Health Sciences, Kawasaki-ku, Kawasaki 210-9501, Japan; mogami@nihs.go.jp (T.N.-M.); tamehiro@nihs.go.jp (N.T.); n-shibata@nihs.go.jp (N.S.); 4Department of Pharmacy, Faculty of Pharmacy, Gifu University of Medical Science, 4-3-3 Nijigaoka, Kani 509-0293, Japan; hmandai@u-gifu-ms.ac.jp

**Keywords:** anti-aging, melanin-producing cells, cytotoxic effects, *ortho*-quinone, pterostilbene, resveratrol

## Abstract

Pterostilbene (PTS), which is abundant in blueberries, is a dimethyl derivative of the natural polyphenol resveratrol (RES). Several plant species, including peanuts and grapes, also produce PTS. Although RES has a wide range of health benefits, including anti-cancer properties, PTS has a robust pharmacological profile that includes a better intestinal absorption and an increased hepatic stability compared to RES. Indeed, PTS has a higher bioavailability and a lower toxicity compared to other stilbenes, making it an attractive drug candidate for the treatment of various diseases, including diabetes, cancer, cardiovascular disease, neurodegenerative disorders, and aging. We previously reported that RES serves as a substrate for tyrosinase, producing an *o*-quinone metabolite that is highly cytotoxic to melanocytes. The present study investigated whether PTS may also be metabolized by tyrosinase, similarly to RES. PTS was oxidized as a substrate by tyrosinase to form an *o*-quinone, which reacted with thiols, such as *N*-acetyl-L-cysteine, to form di- and tri-adducts. We also confirmed that PTS was taken up and metabolized by human tyrosinase-expressing 293T cells in amounts several times greater than RES. In addition, PTS showed a tyrosinase-dependent cytotoxicity against B16BL6 melanoma cells that was stronger than RES and also inhibited the formation of melanin in B16BL6 melanoma cells and in the culture medium. These results suggest that the two methyl groups of PTS, which are lipophilic, increase its membrane permeability, making it easier to bind to intracellular proteins, and may therefore be more cytotoxic to melanin-producing cells.

## 1. Introduction

Stilbenoids are a group of polyphenols found widely in various fruits, aromatic oils, and edible plants. Research studies have shown that stilbenoids have a wide range of biological activities, including anti-cancer, lipid-lowering, anti-diabetic, and cancer chemopreventive activities [1]. Resveratrol (3,5,4′-trihydroxy-*trans*-stilbene, designated as RES (**1**) in Figure 1), was first isolated in 1940 from the roots of white hellebore (*Veratrum grandiflorum O. Loes*), and later from the roots of *Polygonum grandiflorum*, a plant used in traditional medicines in China and Japan. RES (**1**) is also a naturally occurring polyphenol found in several plants, including blueberries, mulberries, cranberries, peanuts, grapes, and wine [2]. Pterostilbene (3,5-dimethoxy-4′-hydroxy-*trans*-stilbene, designated as PTS (**2**) in Figure 1), a natural methylated analogue of RES (**1**), was first isolated from red sandalwood (*Pterocarpus santalinus*) and is one of the active ingredients of *Pterocarpus marsupium*, which is used in traditional medicines to treat diabetes [3]. PTS (**2**) is found mainly in blueberries, grapes, and the wood of some plants [4]. Both RES (**1**) and PTS (**2**) exhibit a wide range of biological activities, including anti-cancer [5,6,7], anti-oxidant [1,8], anti-inflammatory [9,10], anti-aging [11,12], cardioprotective [13,14], and neuroprotective actions [15], which collectively contribute to the prevention of various chronic human diseases. PTS (**2**) is structurally similar to RES (**1**) except that the 3- and 5-positions of the A-phenyl ring are replaced by methoxy groups. The A-phenyl ring has recently received considerable attention due to its promising chemopreventive and chemotherapeutic properties [16,17,18]. The two methyl groups on the chemical backbone of PTS (**2**) make it more lipophilic, allowing it to be absorbed into cells more easily and at a higher rate than RES (**1**), which only has hydroxyl groups in place of the methyl groups [19,20,21]. Animal studies also have proven that the bioavailability of PTS (**2**) is 3- to 4-fold higher than RES (**1**) [22]. PTS (**2**) was shown to be an effective apoptotic and autophagy agent that can inhibit cancer cell survival, induce cell cycle arrest, alter the expression of apoptosis genes, promote levels of autophagy-related proteins, and inhibit cancer cell metastasis [17,23,24]. Despite the existence of both *trans* and *cis* forms, stilbenes exist mostly in the *trans* form in nature [25], with the entire molecule being a conjugated system and the π electron cloud extending throughout the molecule. In this sense, RES (**1**), which is structurally related to stilbene, exists in two isomers, *cis* and *trans*, of which the *trans* isomer is more effective in modulating estrogen-induced biological responses [6,26] and is more stable than the *cis* isomer at pHs between 1 and 7 [27].

3′-Hydroxypterostilbene (3,5-dimethoxy-3′,4′-dihydroxy-*trans*-stilbene, designated as PTS-catechol (**4**) or 3′-hydroxyPTS (**4**) in Figure 2), one of the metabolites of PTS (**2**) [28], can also be isolated from the herb *Sphaerophysa salsula*, a shrub widely distributed in central Asia and northwest China [29,30]. Recent research studies showed that PTS-catechol (**4**) appears to contribute stronger biological activities than PTS (**2**) in human breast cancer cells [31], and possesses anti-adipogenic, anti-inflammatory, anti-oxidant, and sirtuin 1 (Sirt-1) inhibitory activities [32].

Aging is a major risk factor for noninfectious chronic diseases, such as diabetes, cardiovascular disease, cancer, and neurological disorders [33,34]. It has been suggested that polyphenols are involved in the prevention of age-related diseases including atherosclerosis, cardiovascular disease, cancer, arthritis, cataracts, osteoporosis, type 2 diabetes, hypertension, and Alzheimer’s disease [35,36,37]. A recent study [11] suggested that RES (**1**) and PTS (**2**) may both be candidates as anti-aging agents by modulating the hallmarks of aging, such as oxidative damage, inflammation, telomere attrition, and cellular senescence. The anti-oxidant activity of RES (**1**) depends on the redox properties of the phenolic hydroxyl group and the possibility of electron delocalization via the conjugated structure [38]. The anti-oxidant mechanism of RES (**1**) involves reducing the production of reactive oxygen species (ROS), scavenging free radicals, and stimulating the biosynthesis of endogenous anti-oxidants. The increased anti-oxidant capacity and reduced oxidative stress of RES (**1**) have been reported in vitro, in vivo, and in clinical studies [39,40].

Ito et al. [41] reported that RES (**1**) is a good substrate for tyrosinase, and the *o*-quinone produced by its oxidation rapidly forms oligomers and exhibits pro-oxidant activity. It has been reported that RES (**1**) reduces the expression of melanogenesis-related proteins in melanoma cells, and the topical application of RES (**1**) significantly suppresses hyperpigmentation in guinea pig skin stimulated with ultraviolet B (UVB) in vivo [42]. Meanwhile, some studies have suggested that RES (**1**) may be cytotoxic to melanin-producing cells [43,44]. Further studies are needed to investigate the cytotoxicity of RES (**1**) on melanocytes.

Similar to RES (**1**), there is evidence demonstrating that PTS (**2**) also acts as an anti-oxidant by reducing oxidative stress and enhancing endogenous anti-oxidant levels. First, PTS (**2**) acts as an anti-oxidant against various free radicals, such as hydroxyl, superoxide, and hydrogen peroxide, in a concentration-dependent manner [45]. In addition, treatment with PTS (**2**) attenuates glutamate-induced oxidative stress injury [46] and high glucose-induced oxidative injury by increasing the activation of superoxide dismutase (SOD) and glutathione (GSH), as well as decreasing ROS production via the nuclear factor erythroid 2-related factor 2 (Nrf2) signaling pathway in neuronal cells [47]. PTS (**2**) also exerts protective effects against UVB-induced skin damage and carcinogenesis by maintaining the skin’s anti-oxidant defenses, including GSH levels, catalase, superoxide, and GSH peroxidase activity [48]. Furthermore, PTS (**2**) reduces hepatic oxidative stress in a fructose-induced Type 2 diabetes rat model [49].

Since PTS (**2**) has the same *p*-substituted phenol structure as RES (**1**), it is expected to have similar chemical and biochemical properties as RES (**1**). However, it has not yet been reported whether PTS (**2**) can function as a substrate for tyrosinase, similar to RES (**1**). We performed a comparative study to determine whether PTS (**2**) undergoes the same metabolism as RES (**1**). In addition, 4-(3′,5′-dimethoxy-*trans*-styrenyl)-1,2-benzoquinone (PTS-quinone (**3**)), which is generated by tyrosinase oxidation, can be identified after reduction to the corresponding catechol, PTS-catechol (**4**), with NaBH_4_ or ascorbic acid (AA), as was performed to identify the *o*-quinone of RES [41]. PTS-quinone (**3**) is expected to react with *N*-acetyl-L-cysteine (NAC), L-cysteine (CySH) or GSH through the CySH residue to form adducts at three positions (2′, 5′, and 6′) at the B-phenyl ring of PTS (**2**), since the resorcinol dimethyl ether ring (A ring) with the two methoxy groups is inactive to tyrosinase oxidation (Figure 2). In this study, we investigated whether PTS (**2**) acts as a substrate for tyrosinase to generate a toxic *o*-quinone (PTS-quinone (**3**)) that binds to the small thiol compound. PTS (**2**) metabolites, compared to RES (**1**) metabolites, were measured in human tyrosinase-expressing 293T cells (hTYR-293 cells) and in their culture medium. Furthermore, the tyrosinase-dependent cytotoxicity in B16BL6 melanoma cells was also examined. Although the inhibition of melanin synthesis by RES (**1**) has already been reported [42,50], we investigated whether PTS (**2**) and RES (**1**) suppress the synthesis of eumelanin and pheomelanin in B16BL6 melanoma cells and their culture medium, and which one exhibits a stronger inhibitory effect on melanin production.

## 2. Results

### 2.1. Tyrosinase-Catalyzed Oxidation of PTS (***2***)

Similar to RES (**1**) [41], the oxidation of 100 µM PTS (**2**) by mushroom tyrosinase (50 U/mL) was carried out at 37 °C in 50 mM sodium phosphate buffer at pH 6.8. UV/visible spectral changes were followed for 60 min, which showed the rapid production of a quinoid chromophore with absorption at around 450–500 nm, which decayed to a featureless, melanin-like chromophore of PTS-oligomer after 30 min (Figure 3a). Because the changes in the UV/visible spectrum appeared to be complex, we followed the course of oxidation by high-performance liquid chromatography (HPLC). For HPLC analysis, the quinonoid products were reduced to more stable catechols with NaBH_4_. As shown in Figure 3b, the oxidation proceeded slowly at first but then began to proceed rapidly. PTS (**2**) was consumed within 15 min, giving a new compound with a retention time of 5.6 min in HPLC at 45 °C at a flow rate of 1.0 mL/min (PTS (**2**) appeared at 8.1 min), whose structure was assumed to be PTS-catechol (**4**) (3′-hydroxyPTS (**4**)). The structure of this PTS-catechol (**4**) was identified by comparison with commercially available 3′-hydroxyPTS (**4**) by ^1^H-NMR. This identification of PTS-catechol (**4**) led to confirmation of the structure of the immediate product with an absorption maximum at 475 nm as PTS-quinone (**3**).

The production of PTS-catechol (**4**) during the tyrosinase-catalyzed oxidation was further confirmed by oxidizing 100 µM PTS (**2**) in the presence of 10 mol eq. ascorbic acid (AA, 1000 µM) at pH 6.8. As shown in Figure 4a, no quinone formation was observed in the UV-visible spectra changes of PTS (**2**). These results (Figure 4a) indicate that the resulting PTS-quinone (**3**) was immediately reduced by the reducing agent AA to give PTS-catechol (**4**), with an absorption peak at 318 nm (the absorption peak of the starting material, PTS (**2**), is 306 nm). HPLC analysis of the reaction mixture showed that PTS (**2**) was rapidly consumed within 2 min to give PTS-catechol (**4**) in the presence of AA (Figure 4b). The production of PTS-catechol (**4**) decreased rapidly after 5 min. The reason for this decrease will be discussed later.

### 2.2. Reaction of PTS-Quinone (***3***) with Non-Protein Thiol Compounds Producing Mono- and Di-Adducts

We then investigated whether PTS-quinone (**3**) could bind to thiol compounds. NAC was selected as a model of biologically important representative thiol compounds. The merit of using NAC is that the adducts can be extracted with organic solvents, such as ethyl acetate [51]. One hundred µM PTS (**2**) was oxidized by tyrosinase (50 U/mL) in the presence of 200 µM or 300 µM NAC at pH 6.8. When PTS (**2**) was oxidized by tyrosinase in the presence of 2 mol eq. NAC at pH 6.8, the oxidation proceeded rapidly to produce an *o*-quinone pigment with an absorption maximum at approximately 450 nm within 10 min (Figure 5a). Usually, the oxidation of phenols (and catechols) by tyrosinase in the presence of 2 mol eq. thiol produces colorless monothiol adducts as the major product [52,53], but this was not the case with PTS (**2**), as observed for RES (**1**) [41]. We therefore increased the NAC concentration to 3 mol eq. (300 µM). No quinone formation was observed, but instead two absorption maxima at 286 and 322 nm became more distinct than in the case of 2 mol eq. NAC (Figure 5b). When 2 eq. NAC was used, HPLC analysis at a 2-min reaction time showed the production of two major compounds having retention times of 5.6 and 7.3 min for the di-adduct and the mono-adduct in HPLC at 45 °C at a flow rate of 0.7 mL/min, respectively (the starting material PTS (**2**) appeared at 11.1 min). We could not isolate those mono-adduct compounds and therefore the assignment of the mono-adduct is tentative. When 3 eq. NAC was used, a new tri-adduct appeared immediately at a retention time of 4.0 min on the HPLC (Figure 6a). The tri-adduct production became constant after a 5-min reaction time along with the di-adduct, whereas what was expected to be the mono-adduct disappeared quickly (Figure 6b).

To isolate those NAC-adducts, tyrosinase-catalyzed oxidation was performed on a preparative scale. Tyrosinase-catalyzed oxidation of 900 µM PTS (**2**) and 4 mM NAC was performed in the presence of 100 µM PTS-catechol (**4**) as a catalyst, after which the reaction products were isolated by preparative HPLC to give two NAC adducts. After separation, structural analysis of the isolated compounds was performed. ^1^H-NMR analysis of the product with the shorter retention time of 4.0 min on analytical HPLC (Figure 6a) gave three *meta*-oriented protons on the A ring, two protons on the *trans*-oriented double bond, and three NAC moieties (Appendix A). ^13^C NMR analysis (Appendix A) gave signals consistent with the TriNAC-PTS-catechol (**6**). High-resolution MS analysis of this compound gave pseudo-molecular ion peaks at *m*/*z* 754.14381 ([M − H]^−^). These assignments led to the structure of TriNAC adduct (**6**). ^1^H NMR analysis of the next product with the longer retention time (5.6 min) gave signals for one isolated proton on the catechol ring (B ring), three *meta*-oriented protons on the A ring, two protons on the *trans*-oriented double bond, and two NAC moieties (Appendix A). ^13^C NMR analysis (Appendix A) gave signals consistent with the DiNAC-PTS-catechol (**5**). PTS-catechol (**4**) was identified by comparison of its ^1^H-, ^13^C-NMR spectrum, and high-resolution MS analysis with commercially available PTS-catechol (**4**) and its HPLC retention time of 7.8 min (Figure 6a, Appendix A). The location of the protons on the B ring was also confirmed by conventional 2D NMR analysis (Double Quantum Filtered Correlation Spectroscopy (DQF-COSY), Hetero-nuclear Multiple Bond Connectivity (HMBC), and Heteronuclear Multiple Quantum Correlation (HMQC), (Appendix A)). The NMR assignments of DiNAC-PTS-catechol (**5**) and TriNAC-PTS-catechol (**6**) were also referenced to the NMR spectra of DiNAC-RES-catechol and TriNAC-RES-catechol [41].

The isolated yields of DiNAC-PTS-catechol (**5**) and TriNAC-PTS-catechol (**6**) were 27% and 2.9%, respectively, after separation by preparative HPLC. We suggest that the low yield of the tri-adduct is due to its susceptibility to oxidation during isolation. The relatively high yields of the DiNAC adduct, and the low yield of the mono-adduct in analytical HPLC (Figure 6a) are surprising when compared to other reactions of *o*-quinones with cysteine [52]. The extremely high reactivities of PTS-quinone (**3**) and NAC-substituted PTS-quinones must be attributed to the presence of conjugated styrene groups. This was also observed in the similar reactivity of RES-quinone [41]. It has been reported that the tyrosinase-catalyzed reaction of chlorogenic acid, which has a conjugated double bond connected to the catechol ring, results in the production of a tri GSH adduct in the presence of excess GSH [53].

### 2.3. Metabolism of PTS (***2***) and RES (***1***) in Human Tyrosinase-Expressing 293T Cells (hTYR-293T Cells)

We next investigated whether PTS (**2**) is indeed oxidized to PTS-quinone (**3**) in cells expressing human tyrosinase and whether PTS-quinone (**3**) subsequently binds to cellular thiols. We previously developed a cell-based method for metabolite analysis using human tyrosinase-expressing 293T cells (hTYR-293T cells), in which leukemia-inducing phenolic compounds were efficiently converted to their *o*-quinone metabolites [54]. hTYR-293T cells were exposed to PTS (**2**) and RES (**1**) for 2 h. The 100 µM concentration of PTS (**2**) and RES (**1**) is the maximum non-toxic dose (Appendix A). The expression of tyrosinase in these cells was indicated by a marked reduction in viability upon exposure to the anti-melanoma agent 4-*S*-cysteaminylphenol (4SCAP, 300 μM), while there was no reduced viability in mock-transfected cells (Appendix A). 4SCAP has been reported to show tyrosinase-dependent cytotoxicity and to cause a marked decrease in cell viability in hTYR-293T and in B16BL6 melanoma cells [54]. As shown in Figure 7a,b, RES (**1**) and PTS (**2**) were taken up by hTYR-293T cells and were metabolized to Cys-RES-catechol, GS-RES-catechol, Cys-PTS-catechol, and GS-PTS-catechol, respectively, in a dose-dependent manner. The amounts of these Cys and GSH catechol adducts formed in cells and excreted into the medium were calculated as the sum of two mono-adducts and one di-adduct, as has been reported for RES (**1**) [41]. The RES (**1**) content in cells exposed to 100 µM RES (**1**) was 1.65 nmol/million cells, which contained 0.037 nmol/million cells Cys-RES-catechol and 0.215 nmol/million cells GS-RES-catechol (Figure 7a). On the other hand, the PTS (**2**) content in cells exposed to 100 µM PTS (**2**) was 14.7 nmol/million cells, which contained 0.078 nmol/million cells Cys-PTS-catechol and 0.292 nmol/million cells GS-PTS-catechol. These metabolites were then excreted into the medium (Figure 7c,d). The RES (**1**) content in medium exposed to 100 µM RES (**1**) was 65 µM, which contained 0.039 µM Cys-RES-catechol and 0.006 µM GS-RES-catechol. On the other hand, the PTS (**2**) content in medium exposed to 100 µM PTS (**2**) was 59 µM, which contained 0.016 µM Cys-PTS-catechol and 0.088 µM GS-PTS-catechol.

When cells were exposed to 100 µM PTS (**2**), the intracellular concentration of PTS (**2**) was 8.9 times greater than the RES (**1**) content in cells exposed to 100 µM RES (**1**). The intracellular contents of Cys-PTS-catechol and GS-PTS-catechol in cells exposed to 100 μM PTS (**2**) were also 2.1 times and 1.4 times higher than the RES (**1**) contents in cells exposed to 100 μM RES (**1**), respectively. The metabolites in the cells and medium in PTS (**2**) were several times higher than in RES (**1**). These results indicate that PTS (**2**) is more easily taken up by cells than RES (**1**) and is excreted in larger amounts into the medium. The fact that PTS (**2**) was taken up into cells in large quantities and then excreted into the medium is in parallel with the fact that PTS (**2**) is much more hydrophobic than RES (**1**) based on its structure.

### 2.4. Evaluation of Tyrosinase-Dependent Cytotoxicity in B16BL6 Melanoma Cells

We next investigated the tyrosinase-dependent cytotoxicity by knocking tyrosinase down with siRNA to inhibit it. B16BL6 melanoma cells were transfected with a negative-control siRNA or with a siRNA directed against tyrosinase for 24 h, which effectively diminished the expression level of tyrosinase mRNA (Figure 8a). The cells were treated with the indicated concentrations of 4SCAP, RES (**1**) and PTS (**2**) for 24 h, and their viability was measured. As shown in a previous study [54], a tyrosinase-dependent enhancement of cytotoxicity was observed with 4SCAP (Figure 8b). The knockdown of tyrosinase suppressed the cytotoxicity induced by 24 h treatment with higher concentrations of RES (**1**) and PTS (**2**) compared to the negative control siRNA-transfected B16BL6 melanoma cells (Figure 8c,d). Regarding cytotoxicity, there were no significant differences between RES (**1**) and PTS (**2**) at lower concentrations, but PTS (**2**) was slightly but significantly more cytotoxic against B16BL6 melanoma cells than RES (**1**) at a higher concentration (100 µM). The *p* values of siRNA Ctrl and siRNA *Tyr* between RES (**1**) and PTS (**2**) at 100 µM were 0.017 and 0.020, respectively. Both compounds contributed significantly to the tyrosinase-dependent cytotoxicity.

### 2.5. Inhibition of Melanin Synthesis by PTS (***2***) and RES (***1***) in B16BL6 Melanoma Cells

It is reported that RES (**1**) potently inhibits melanin synthesis in murine melanoma B16/F10 cells and human epidermal melanocytes at a concentration as low as 3 µM [55]. Therefore, we next investigated whether PTS (**2**) also inhibits melanogenesis in melanoma cells. After B16BL6 melanoma cells were cultured with PTS (**2**) or RES (**1**) for 24 h, the cultures were separated into cells and medium and were then analyzed for changes in eumelanin and pheomelanin contents. Eumelanin and pheomelanin can be calculated by multiplying the amounts of pyrrole-2,3,5-tricarboxylic acid (PTCA) after alkaline hydrogen peroxide oxidation (AHPO) and 4-amino-3-hydroxyphenylalanine (4-AHP) after hydroiodic acid (HI) hydrolysis by factors of 25 and 9, respectively [56,57,58]. PTS (**2**) and RES (**1**) significantly inhibited melanin production at 30 µM in cells (Figure 9a,b). Surprisingly, RES (**1**) promoted the production of eumelanin at 3 and 10 µM, while pheomelanin was decreased in a concentration-dependent manner by both PTS (**2**) and RES (**1**). In the medium, both PTS (**2**) and RES (**1**) inhibited eumelanin and pheomelanin levels (excretion) even at 3 µM, and this effect was more pronounced for pheomelanin (Figure 9c,d). At concentrations of 10 and 30 µM in the medium, PTS (**2**) showed similar degrees of excretion of eumelanin and pheomelanin as RES (**1**), but PTS (**2**) was more effective in inhibiting pheomelanin levels at 3 µM (*p* value = 0.002; 0.026 for eumelanin synthesis).

## 3. Discussion

Phenols/catechols with aliphatic/aromatic side chains at the *para* position (4-position) are representative of compounds that cause leukoderma in humans and in animals [59,60]. Compounds belonging to this class, such as 4-*tert*-butylphenol (4-TBP), hydroquinone monobenzyl ether (MBEH), *p*-cresol (CRE), and raspberry ketone (RK), which are known to cause occupational leukoderma, and the anti-melanoma drug candidates 4SCAP, its derivatives *N*-acetyl-4SCAP (NAc-4SCAP) and *N*-propionyl-4SCAP (NPr-4SCAP), have been reported to exert selective toxicity against melanoma cells and follicular melanocytes [51,61,62,63,64]. The chemical structure of rhododendrol (rhododenol, RD) also belongs to this class. These leukoderma-inducing phenols have been reported to be commonly metabolized by tyrosinase to generate highly toxic *o*-quinones. There has been an active development of cosmetic ingredients that inhibit melanin production, so-called “skin whitening” ingredients. RD was approved as a quasi-drug ingredient that inhibits melanin production by competitively inhibiting tyrosinase and was sold commercially for six years. However, it was later found that users of medicated cosmetics containing RD frequently developed “leukoderma” at the skin application site, and the product was voluntarily recalled in 2013. RD was developed as an inhibitor of melanin production due to its inhibition of tyrosinase [65,66], but it has been shown to be metabolized by tyrosinase and converted into active metabolites that lead to cytotoxicity. We [67] have shown that RD is metabolized by tyrosinase and activated to an *o*-quinone, like the leukoderma-inducing 4-substituted phenols, 4-TBP, MBEH, CRE, RK, and 4SCAP, and undergoes an addition reaction with SH compounds such as CySH and GSH [54], and that the metabolic activation to *o*-quinone also proceeds efficiently using human tyrosinase [68]. We reported the effect of tyrosinase knockdown using siRNA on the cytotoxicity of RD, MBEH, and 4-TBP in mouse B16BL6 melanoma cells, and found that some leukoderma-inducing phenols exhibited tyrosinase-dependent cytotoxicity [54].

In this study, we used hTYR-293T cells to generate high expression levels of human tyrosinase. Using this cell model, we investigated whether the generation of *o*-quinones leads to the production of Cys- and GSH catechol adducts. We also tested tyrosinase-dependent cytotoxicity using B16BL6 melanoma cells. Although *p*-substituted phenols may exhibit potential cytotoxicity to melanocytes, the anti-pigmentation effect of RES (**1**) has been also investigated [42]. Some studies suggest that RES (**1**) may be cytotoxic to melanocytes. Evidence suggesting that RES (**1**) is cytotoxic to melanocytes was also reported by a study demonstrating that RES-quinone can bind to bovine serum albumin through its cysteine residues and that RES-oligomers can oxidize GSH to GSSG, leading to its pro-oxidant activity [41]. Due to the high reactivity of thiol groups, proteins are able to react with *o*-quinones [69]. This type of protein modification is biologically important because it can lead to the denaturation of thiol proteins and the inhibition of thiol enzymes, resulting in cytotoxicity [69,70,71]. RES (**1**) strongly inhibited the proliferation of HT-144 human melanoma cells with an IC_50_ of 5–10 μM [43]. Similarly, RES (**1**) inhibited the proliferation of B16F1 mouse melanoma cells and NHEMb human epidermal melanocytes with IC_50_s of 27.1 μM and 38.2 μM, respectively. This is comparable to the values for hydroquinone (28.3 and 9.4 μM) [44]. Meanwhile, another study found that the cytotoxicity of RES (**1**) was rather weak, with an IC_50_ of less than 100 μM [42], so further studies are clearly needed to investigate the cytotoxicity of RES (**1**) to melanocytes. These results indicate that the use of RES (**1**) in cosmetics should be considered with caution. RES (**1**) is generally considered to be a good inhibitor rather than a substrate for tyrosinase [55,72]. However, Ito et al. [41] showed that RES (**1**) is also a good substrate for mushroom tyrosinase, producing the reactive *o*-quinone, RES-quinone. These situations are similar to those observed when RD is both a good inhibitor and a good substrate for tyrosinase [65,66,67], including human tyrosinase [68].

The effect of PTS (**2**) on cell viability, as assessed using the MTT assay [73], showed that no cytotoxicity was observed at concentrations below 3 μM, and melanin content and tyrosinase activity were decreased in a dose-dependent manner after treatment with PTS (**2**). Although PTS (**2**) has many applications, including anti-inflammatory, anti-oxidant, and antitumor effects, its skin whitening effect has recently attracted more and more attention. However, the mechanisms of inhibition of melanin production and melanosome transport still require further study [73]. Compared with 4SCAP, which caused a strong tyrosinase-dependent decrease in cell viability in B16BL6 melanoma cells, PTS (**2**) and RES (**1**) showed a weaker cytotoxicity. Nevertheless, both compounds contribute significantly to the tyrosinase-dependent cytotoxicity, and PTS (**2**) showed a slightly stronger cytotoxicity than RES (**1**) at a concentration of 100 µM (Figure 8).

PTS (**2**) mainly plays a role by inhibiting the cAMP/PKA/CREB signal pathway. After the cAMP/PKA/CREB signal pathway is inhibited, tyrosinase activity and the expression levels of MITF, TYR, Rab27A, Rab17, and gp100 decreased, and melanin production, melanocyte dendrite development, and melanosome transport were suppressed [73].

Although several studies have reported that RES (**1**) can inhibit melanin production [42,50], the whitening effect of PTS (**2**) has not been elucidated. Meanwhile, the anti-melanogenic activity of PTS (**2**) was examined in UVB-irradiated B164A5 mouse melanoma cells, and it was reported that 5 or 10 µM PTS (**2**) inhibited tyrosinase activity in a dose-dependent manner by 37% or 58%, respectively, and that PTS (**2**) had a 10-fold stronger anti-melanogenic activity than RES (**1**) [74]. RES (**1**) reduced the expression of melanogenesis-related proteins, such as tyrosinase, tyrosinase-related protein 1 (TYRP1), TYRP2, and MITF in melanoma cells [42]. It has been reported that PTS (**2**) is a more potent melanogenesis inhibitor than RES (**1**) in α-melanocyte-stimulating hormone-stimulated B16F10 melanoma cells, and PTS (**2**) inhibited tyrosinase activity in a dose-dependent manner [75]. According to some studies [76], PTS (**2**) can inhibit UVA-mediated pigmentation by inducing autophagy in melanocytes. It has also been reported that PTS (**2**) exhibits a strong inhibitory effect on melanin production by B16F10 cells (3 µM), by human skin in vitro (10 µM), and by zebrafish embryos (3 µM), and inhibits melanocyte dendritic development and melanosome transport [73].

However, the chemopreventive effects of PTS (**2**) via its anti-tyrosinase activity and inhibitory effects on melanin content have not been reported previously. In this study, we demonstrated that tyrosinase can effectively oxidize PTS (**2**) and RES (**1**) to generate *o*-quinone skeletons (**3**), the high reactivity of which is demonstrated by their rapid decay. When PTS (**2**) was oxidized by tyrosinase in the presence of 3 eq. NAC, it yielded the Di- and Tri-NAC adducts similar to RES (**1**), which remained stable [41]. We investigated whether PTS (**2**) and RES (**1**) could be oxidized to their respective quinones to generate Cys- and GS-catechol derivatives in hTYR-293T cells (Figure 7). PTS (**2**) and RES (**1**) were metabolized to Cys- and GS-catechol metabolites in a dose-dependent manner and were excreted into the culture medium. The intracellular concentration of PTS (**2**) was 8.9-fold higher than RES (**1**). The amounts of Cys-PTS-catechol and GS-PTS-catechol in the cells were also approximately 2.1- and 1.4-fold higher than the amounts of RES (**1**), respectively. The amounts of Cys-PTS-catechol and GS-PTS-catechol in the culture medium were several-fold higher than those of RES (**1**). These results indicated that PTS (**2**) was more easily taken up by cells than RES (**1**) and was more significantly excreted into the medium as CySH and GSH adducts (Figure 7). The formation of Cys- and GS-catechols indicated that PTS-quinone (**3**) or RES-quinone, which are cytotoxic to cells, were trapped as thiol adducts, which may lead to cellular CySH and GSH reduction. Both RES (**1**) and PTS (**2**) serve as substrates for tyrosinase and were rapidly consumed by treated cells to form catechol adducts and then released into the cell medium. The amounts and nature of the melanins that accumulated in the cells and extracellular medium were analyzed, and both RES (**1**) and PTS (**2**) inhibited the production of eumelanin and pheomelanin by ca. 50% in B16BL6 melanoma cells at a low concentration of 30 µM (Figure 9). When the effect on melanin synthesis was examined in the culture medium, even 3 µM PTS (**2**) inhibited the production of eumelanin and pheomelanin by 34% and 55%, respectively, which were significantly greater than RES (**1**).

RD was previously reported to inhibit melanin production in the skin through a mechanism believed to be the competitive inhibition of tyrosinase, and its 4-alkylphenol structure suggests that RD may also be a good substrate for tyrosinase producing the cytotoxic *o*-quinone [64,67]. The present results suggest that RES (**1**) and PTS (**2**), especially the latter, also serve as substrates for tyrosinase, leading to cytotoxicity of melanoma cells at higher concentrations, although at lower concentrations RES (**1**) and PTS (**2**) may inhibit melanin production through the inhibition of tyrosinase activity.

Lastly, in the tyrosinase oxidation of PTS (**2**) in the presence of AA (Figure 4b), PTS-catechol (**4**) rapidly decreased after 5 min of oxidation. In this connection, we have reported that the tyrosinase oxidation of RK produces 3,4-dihydroxybenzalacetone (DBL)-quinone, which undergoes an ionic Diels–Alder condensation producing dimers and trimers [77]. It is possible that, similar to the DBL reaction, the Diels–Alder reaction of PTS-catechol (**4**) with PTS-quinone (**3**) produces the dimers, which are further oxidized to produce PTS (**2**)-oligomers.

## 4. Materials and Methods

### 4.1. Materials

Pterostilbene (3,5-dimethoxy-4′-hydroxy-trans-stilbene) (PTS, **2**), resveratrol (RES, **1**), and piceatannol (RES-catechol), were purchased from Tokyo Chemical Industries (Tokyo, Japan). 3′-Hydroxypterostilbene (PTS-catechol, **4**) was purchased from Selleck Chemicals (Houston, TX, USA). Tyrosinase (from mushrooms, specific activity 2687 U/mg), L-cysteine (CySH), reduced glutathione (GSH), oxidized glutathione (GSSG), *N*-acetyl-L-cysteine (NAC), and Dulbecco’s Modified Eagle’s Medium (DMEM) containing 10% fetal bovine serum were purchased from Sigma-Aldrich (St. Louis, MO, USA). Perchloric acid (HClO_4_) was purchased from Katayama Chemical Industries Co., Ltd. (Osaka, Japan). Formic acid, NaBH_4_, and methanol (HPLC grade) were from FUJIFILM Wako Pure Chemical Corporation (Osaka, Japan). The highest purity Milli-Q water (Milli-Q Advantage, Merck Millipore Co., Tokyo, Japan) was used throughout this study to avoid contamination with metal ions.

### 4.2. Instruments

An HPLC system was used to follow the course of tyrosinase oxidation. It was comprised of an analytical UV/VIS detector, a JASCO pump (JASCO Co., Tokyo, Japan), a C18 column (Capcell Pak MG; 4.6 × 250 mm; 5 µm particle size, Osaka Soda, Osaka, Japan), and a JASCO UV/visible detector (JASCO Co., Tokyo, Japan). The mobile phase was 0.4 M formic acid/methanol, 30:70 (*v*/*v*), 70:30 (*v*/*v*) or 50:50 (*v*/*v*). Analyses were performed at 45 °C or 40 °C at a flow rate of 0.7 mL/min or 1.0 mL/min. For preparative separation, a C18 preparative column (Capcell Pak MG; 20 × 250 mm; 5 µm particle size, Osaka Soda, Osaka, Japan) was used at a flow rate of 7.0 mL/min with the mobile phase (0.4 M formic acid/methanol, 30:70 (*v*/*v*)) at 40 °C. A Shiseido electrochemical detector (Shiseido, Tokyo, Japan) was also used to analyze low concentrations of metabolites, as electrochemical detection is >10-times more sensitive than UV/visible detection. Nuclear magnetic resonance (NMR, 400 MHz for ^1^H) spectra were obtained using a JNM-ECZ400S spectrometer (JEOL Ltd., Tokyo, Japan). ^1^H NMR (400 MHz) and ^13^C NMR (100 MHz) spectra were obtained in CD_3_OD. Chemical shifts were referenced to the solvent signals (3.30 ppm for ^1^H and 49.0 ppm for ^13^C). High-resolution mass spectra were obtained using a 6220 TOF mass spectrometer (mode: electrospray ionization—time-of-flight, negative; ESI(−) (Shimazu LCMS-9030, Q-TOF/HPLC (Nexera), Kyoto, Japan).

### 4.3. Oxidation of PTS (***2***) by Tyrosinase in the Absence or Presence of L-Ascorbic Acid or NAC

According to the method described by Ito et al. [41], a solution containing about 8% ethanol (2 mL) of 100 µM PTS (**2**) was oxidized by 50 U/mL tyrosinase at 37 °C in 50 mM sodium phosphate buffer (pH 6.8). Changes in absorption spectra were periodically followed for 60 min. The oxidation was also carried out in the presence of 1 mM AA, 200 µM NAC or 300 µM NAC. By adding a small amount of ethanol, it became possible to obtain reproducible experimental results with PTS (**2**), which has a high hydrophobicity. For spectrophotometric analysis, the reference cell contained the same concentrations of buffer, tyrosinase, ethanol, and AA or NAC. For HPLC analysis, the reaction was stopped by adding a 200 µL aliquot to 20 µL 10% NaBH_4_ followed by 180 µL 0.8 M HClO_4_, and 400 µL methanol. The addition of methanol suppressed the adsorption of PTS (**2**) and its metabolites in reaction tubes in vitro, and on HPLC columns.

### 4.4. Isolation of DiNAC-PTS-Catechol (***5***) and TriNAC-PTS-Catechol (***6***)

According to the method described by Ito et al. [41], a solution of PTS (**2**) (23.1 mg, 90 µmol) and PTS-catechol (**4**) (2.7 mg, 10 µmol), and NAC (65.2 mg, 400 µmol) in ethanol (10 mL) was mixed with 92 mL 50 mM sodium phosphate buffer (pH 6.8). Catalytic PTS-catechol (**4**) was added to promote the reaction [41]. The mixture was vigorously shaken at 37 °C to which tyrosinase (20,000 U) in 1 mL buffer was added. After 30 min of oxidation when the PTS (**2**) was consumed by 50%, 2 mL 6 M HCl was added to stop the oxidation, and the mixture was extracted twice with 100 mL ethyl acetate. After evaporation of ethyl acetate under reduced pressure, the residue was dissolved in HPLC elution buffer and was subjected to preparative HPLC. After drying each eluate, 16 mg (27% yield) of DiNAC-PTS-catechol (**5**) (HPLC purity 90%), 2.0 mg (2.9%) of TriNAC-PTS-catechol (**6**), (HPLC purity 100%), and 2.0 mg (HPLC purity 100%) of PTS-catechol (**4**) were obtained.

#### 4.4.1. 3,5-Dimethoxy-3′,4′-hydroxy-*trans*-stilbene (PTS-catechol (**4**) or 3′-hydroxyPTS (**4**))

The ^1^H- and ^13^C-NMR spectra of PTS-catechol (**4**) are shown in Appendix A. the ^1^H NMR spectrum with the 2D DQF-COSY method is shown in Appendix A. The ^1^H and ^13^C NMR spectra with 2D HMBC, and the HMQC method is shown in Appendix A, respectively. High-resolution MS, *m*/*z* 271.0983 ([M − H]^−^), calc’d for C_16_H_16_O_4_, 272.

#### 4.4.2. 2′,5′-Di-S-[(N-acetyl)cysteinyl]-3,5-dimethoxy-3′4′-dihydroxy-trans-stilbene (DiNAC-PTS-catechol (**5**))

The ^1^H NMR spectrum of DiNAC-PTS-catechol (**5**) is shown in Appendix A, and the ^1^H NMR spectrum with the 2D DQF-COSY method is shown in Appendix A. The ^1^H and ^13^C NMR spectra with 2D HMBC, and the HMQC method is shown in Appendix A, respectively. The ^1^H NMR and ^13^C NMR data are summarized in Appendix A.

#### 4.4.3. 2′,5′,6′-Tri-S-[(N-acetyl)cysteinyl]-3,5-dimethoxy-3′4′-dihydroxy-trans-stilbene (TriNAC-PTS-catechol (**6**))

The ^1^H NMR spectrum of TriNAC-PTS-catechol (**6**) is shown in Appendix A, and the ^1^H NMR spectrum with the 2D DQF-COSY method is shown in Appendix A. The ^1^H and ^13^C NMR spectra with 2D HMBC, and the HMQC method are shown in Appendix A, respectively. The ^1^H NMR and ^13^C NMR data are summarized in Appendix A. High-resolution MS, *m*/*z* 754.14381 ([M − H]^−^), calc’d for C_31_H_37_N_3_O_13_S_3_, 755.

### 4.5. Metabolism of RES (***1***) and PTS (***2***) in Human Tyrosinase-Expressing 293T Cells

Human tyrosinase-expressing 293T (hTYR-293T) cells were prepared by transient transfection as previously described [54]. Briefly, 293T cells (0.75 × 10^6^/well) were seeded in 6-well plates, which contained the expression plasmid for human tyrosinase (1.6 µg) and Lipofectamine 2000 Reagent (4 µg) (Invitrogen, Carlsbad, CA, USA) in 10% fetal bovine serum (FBS)-supplemented DMEM. After 24 h of transfection, the medium (0.75 mL) was replaced with medium containing vehicle (DMSO, 0.5%) with or without 30, 60, and 100 µM RES (**1**) or PTS (**2**). After incubation for 2 h, the cells and medium supernatants were collected. Medium samples (360 µL) were deprotonated with 40 μL 4 M HClO_4_ and 400 μL methanol by shaking vigorously at room temperature for 60 min. Cell samples (0.75 × 10^6^) were extracted with 375 μL 0.4 M HClO_4_ and methanol (1:1) by shaking vigorously at room temperature for 60 min. After centrifuging these treated cells and medium at 10,000 rpm for 3 min, the supernatants were analyzed by HPLC as described below.

Concentrations of RES (**1**) and PTS (**2**) remaining in the medium or taken up in cells were analyzed using a mobile phase of 0.4 M formic acid/methanol = 30:70 (*v*/*v*) at 45 °C with a flow rate of 0.7 mL/min or 1.0 mL/min and a wavelength of 310 nm. RES (**1**) appeared at 4.0 min, and PTS (**2**) appeared at 7.9 min under these conditions. Concentrations of RES-quinone adducts of CySH and GSH in the medium or in cells were analyzed using a mobile phase of 0.4 M formic acid/methanol = 70:30 (*v*/*v*) at 40 °C with a flow rate of 0.7 mL/min. To detect low levels of those adducts, an electrochemical detector with an applied voltage of 600 mV was used in place of the UV detector. Concentrations of PTS-quinone (**3**) adducts of CySH and GSH in the medium or in cells were analyzed using a mobile phase of 0.4 M formic acid/methanol = 50:50 (*v*/*v*) at 45 °C with a flow rate of 0.7 mL/min. To detect low levels of those adducts, an electrochemical detector with an applied voltage of 600 mV was used. Three each of CySH and GSH adducts were produced from RES (**1**) or PTS (**2**). Their levels in cells and culture medium were calculated by comparing the combined peak heights of the adducts with those of the standards (see below).

### 4.6. Adduct Formation of RES- and PTS-Quinones with CySH and GSH

According to the method described by Ito et al. [41], standard GSH- and Cys-adducts of o-quinone metabolites were prepared by tyrosinase-catalyzed oxidation as follows. A solution consisting of 20 µL 10 mM RES catechol or PTS-catechol (**4**) (final 100 µM), 80 µL 10 mM Cys or GSH, 1700 µL 50 mM sodium phosphate buffer, pH 6.8, and 160 μL ethanol was incubated with 20 µL 5000 U/mL mushroom tyrosinase. After shaking for 5 min at 37 °C, the reaction was stopped by the addition of 20 µL 6 M HCl. Subsequently, 2 mL methanol was added and mixed to avoid adsorption of the products. A 10-fold volume of 0.4 M formic acid/methanol = 1:1 (*v*/*v*) was added and mixed, then centrifuged at 10,000 rpm for 3 min, and 180 µL of the supernatant was recovered. The resulting mixtures (equivalent to 2.5 µM combining the three adducts formed) of catechol derivatives from CySH and GSH adducts of PTS (**2**) or RES (**1**) were prepared as standards and were used under the above HPLC conditions. The three Cys- or GS-adducts of RES-catechol or PTS-catechol correspond to two mono-adducts and one di-adduct [41].

### 4.7. Evaluation of Tyrosinase-Dependent Cytotoxicity in B16BL6 Melanoma Cells

B16BL6 mouse melanoma cells were obtained from the RIKEN BioResource Research Center (Tsukuba, Japan) and were maintained in RPMI1640 medium containing 10% FBS. The tyrosinase-dependent cytotoxicity of RES (**1**) or PTS (**2**) was assessed as previously described [54]. For siRNA knockdown of tyrosinase expression, transfection was performed in each separate well of 96-well plates. Cells (1 × 10^4^ cells/well) were transfected with 1 pmol Stealth™ Select RNAi (Thermo Fisher Scientific, Walthem, MA, USA) for mouse *Tyr* (MSS212191) or Stealth RNAi^TM^ negative control using Lipofectamine RNAiMAX reagent (0.2 μL/well) (Thermo Fisher Scientific). Twenty-four h after transfection, the medium was exchanged to growth medium (100 μL) containing vehicle (DMSO, 0.5%) with or without RES (**1**) or PTS (**2**). Cell viability was determined by the CellTiter-Glo^®^ luminescent cell viability assay (Promega Corp., Madison, WI, USA). For the assessment of tyrosinase knockdown, the expression of *Tyr* mRNA in cells was determined 24 h after transfection and then 24 h after incubation in growth medium. Total RNA was extracted using a RNeasy Mini Kit with deoxyribonuclease (QIAGEN, Valencia, CA, USA). Quantitative real-time RT-PCR was performed with an ABI Prism 7300 sequence detection system (Thermo Fisher Scientific) using the QuantiTect Probe RT-PCR (QIAGEN, Valencia, CA, USA). Primer/probe sequences used were as follows: mouse *Tyr*, forward: 5′-AGCCTGTGCCTCCTCTAAGAACT-3′; reverse: 5′-CTTGCCGATGGCCAGAAG-3′; probe: 5′-6-FAM-TTGGCAAAA/ZEN/GAATGCTGCCCACCA-IBFQ-3′ (Integrated DNA Technologies, Coralville, IA, USA) and 18S rRNA (Applied Biosystems, Thermo Fisher Scientific). Expression data were normalized to 18S rRNA levels and are reported as fold differences between control- and *Tyr*-siRNA treated cells.

### 4.8. Evaluation of Tyrosinase-Dependent Cytotoxicity in hTYR-293T Cells

For the assessment of tyrosinase-dependent cytotoxicity, 293T cells (3 × 10^4^ cells/well) were transfected with the expression plasmid for human tyrosinase (80 ng) or with an empty plasmid (80 ng) using Lipofectamine 2000 (0.2 μL) (Thermo Fisher Scientific) in 96-well plates and incubated in DMEM supplemented with 10% FBS for 24 h. The medium was replaced with growth medium (100 μL) containing vehicle (DMSO, 0.5%) with or without RES (**1**), PTS (**2**), or 4SCAP. After a 2-h incubation, cell viability was determined using the CellTiter-Glo^®^ luminescent cell viability assay kit.

### 4.9. Measurement of Melanin Levels in B16BL6 Melanoma Cells and in the Culture Medium

B16BL6 melanoma cells were sonicated in 400 µL Milli-Q water, and 100 µL aliquots were subjected to AHPO and HI hydrolysis followed by PTCA and 4-AHP analyses by HPLC. 100 µL of 4M HClO_4_ were added to 900 µL medium and stored at 4 °C for 1 h. After centrifugation, the sediments were washed with 500 µL 0.4 M HClO_4_, then ultrasonically crushed in 400 µL milli-Q water, and 100 µL of each sample was treated with AHPO and HI hydrolysis followed by PTCA and 4-AHP analyses by HPLC. AHPO and HI hydrolysis were performed according to previously reported methods [56,57].

### 4.10. Statistical Analyses

Student’s *t*-tests (two-tailed) were performed using Microsoft Excel 2019 (version 2408) (Japan Microsoft Co., Tokyo, Japan). A *p* value of <0.05 was considered statistically significant.

## 5. Conclusions

We previously demonstrated that RES (**1**), a naturally occurring polyphenol with anti-oxidant and anti-inflammatory effects, acts as a substrate for tyrosinase to form the *o*-quinone, which is cytotoxic to melanocytes. In this study, we performed a comparative study on the possibility that PTS (**2**), a dimethyl derivative of RES (**1**), undergoes the same metabolism as RES (**1**). Tyrosinase oxidation of PTS (**2**) formed *o*-quinone producing adducts with thiols, such as NAC. The results also confirmed that PTS (**2**) exhibited tyrosinase-dependent cytotoxicity in B16BL6 melanoma cells, and that PTS (**2**) was taken up into cells and was metabolized several times more quickly than RES (**1**) in human tyrosinase-expressing 293T cells. We also confirmed that PTS (**2**) and RES (**1**) can reduce the viability of B16BL6 melanoma cells and suppress melanin synthesis in the cells and in the culture medium. These effects of PTS (**2**) were greater than those of RES (**1**). The present results suggest that both PTS (**2**) and RES (**1**) are substrates for tyrosinase, but at low concentrations they suppress melanin synthesis, which may be due to the inhibition of tyrosinase activity. In conclusion, the results of this study suggest that PTS (**2**) and RES (**1**), especially PTS (**2**), is cytotoxic to melanin-producing cells due to the binding of their *o*-quinones to thiol proteins.

## Figures and Tables

**Figure 1 ijms-25-09990-f001:**
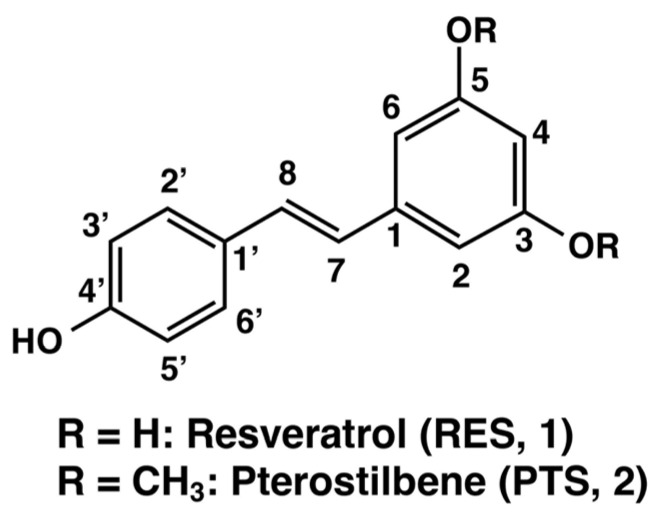
Structures of resveratrol (RES, **1**) and pterostilbene (PTS, **2**).

**Figure 2 ijms-25-09990-f002:**
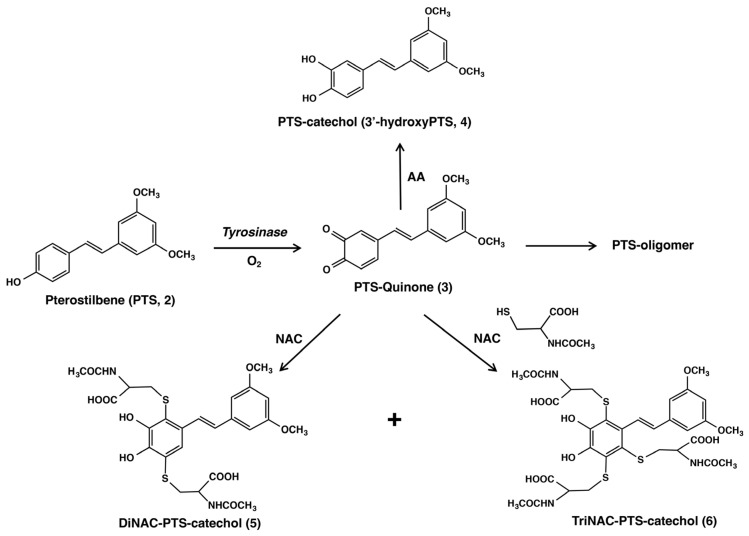
Scheme showing the tyrosinase-catalyzed oxidation of pterostilbene (PTS, **2**) in the absence or presence of a thiol. The oxidation of PTS (**2**) gives PTS-quinone (**3**) as an immediate product, which rapidly decays. PTS-quinone (**3**) is reduced by ascorbic acid (AA) to form PTS-catechol (3′-hydroxyPTS, (**4**)). The tyrosinase-catalyzed oxidation of PTS (**2**) in the presence of the thiol *N*-acetyl-L-cysteine (NAC) affords the di-adduct DiNAC-PTS-catechol (**5**) and the tri-adduct TriNAC-PTS-catechol (**6**). These thiol adducts were isolated and identified as the NAC adducts; “+” means that DiNAC-PTS-catechol (**5**) and TriNAC-PTS-catechol (**6**) are produced together.

**Figure 3 ijms-25-09990-f003:**
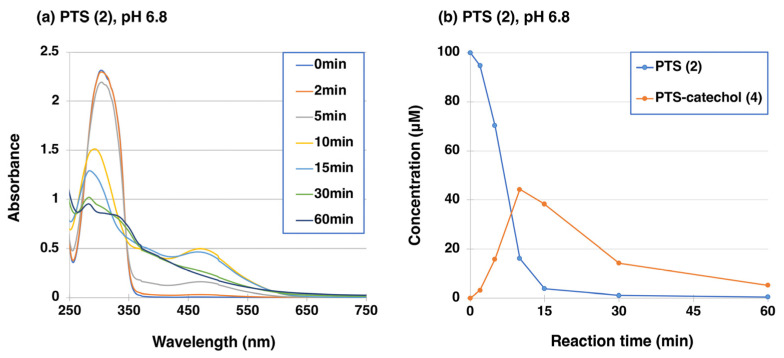
Time course of the tyrosinase-catalyzed oxidation of PTS (**2**) and PTS-catechol (**4**) and HPLC analyses of reaction products. (**a**) UV/visible spectral changes of PTS (**2**) at pH 6.8. (**b**) HPLC analysis following the tyrosinase-catalyzed oxidation of PTS (**2**) at pH 6.8, the reaction being stopped by the addition of NaBH_4_, followed by HClO_4_. The experiments were repeated once, and good reproducibility was obtained. The figure was from a single experiment and is representative.

**Figure 4 ijms-25-09990-f004:**
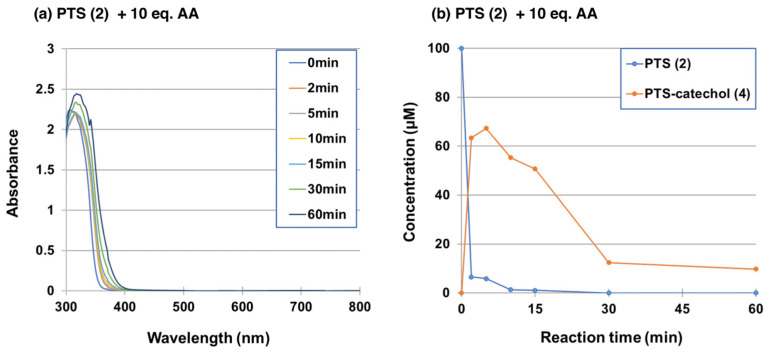
Time course of the tyrosinase-catalyzed oxidation of PTS (**2**) in presence of ascorbic acid (AA). (**a**) UV/visible spectral changes of PTS (**2**) at pH 6.8 in presence of 10 mol eq. AA. (**b**) HPLC analysis following the tyrosinase-catalyzed oxidation of PTS (**2**) at pH 6.8 in presence of 10 mol eq. AA, the reaction being stopped by the addition of NaBH_4_, followed by HClO_4_. The experiments were repeated once, and good reproducibility was obtained. The figure was from a single experiment and is representative.

**Figure 5 ijms-25-09990-f005:**
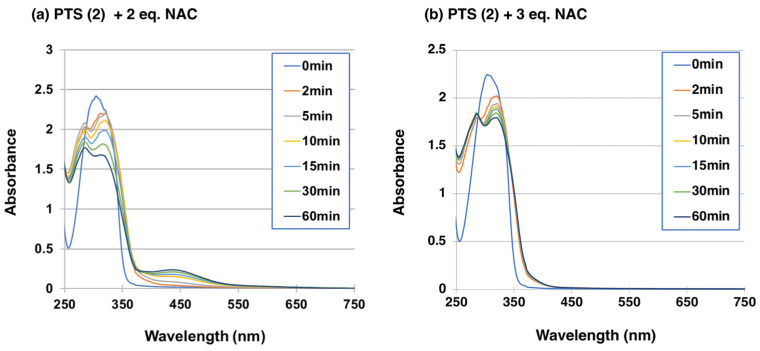
Time course of the tyrosinase-catalyzed oxidation of PTS (**2**) in presence of NAC. (**a**) UV/visible spectral changes of PTS (**2**) in presence of 2 mol eq. NAC at pH 6.8. (**b**) UV/visible spectral changes of PTS (**2**) in presence of 3 mol eq. NAC at pH 6.8. The experiments were repeated once, and good reproducibility was obtained. The figure was from a single experiment and is representative.

**Figure 6 ijms-25-09990-f006:**
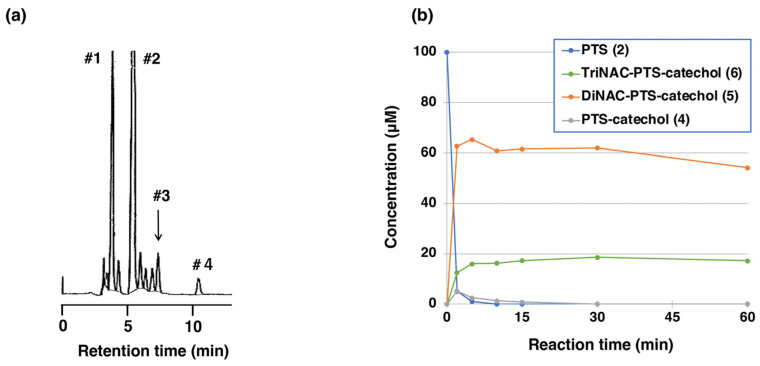
Time course of the tyrosinase-catalyzed oxidation of PTS (**2**) in presence of NAC. (**a**) HPLC chromatogram of the tyrosinase-catalyzed oxidation of PTS (**2**) for 5 min in presence of 3 mol eq. NAC at pH 6.8, the reaction being stopped by the addition of NaBH_4_, followed by HClO_4_. Peak: #1; tri-adduct (**6**), #2; di-adduct (**5**), #3; PTS-catechol (**4**), #4; PTS (**2**). HPLC analyses were performed at 45 °C at a flow rate of 0.7 mL/min. (**b**) HPLC analysis following the tyrosinase-catalyzed oxidation of PTS (**2**) in presence of 3 mol eq. NAC at pH 6.8, the reaction being stopped by the addition of NaBH_4_, followed by HClO_4_. The experiments were repeated once, and good reproducibility was obtained. The figure was from a single experiment and is representative.

**Figure 7 ijms-25-09990-f007:**
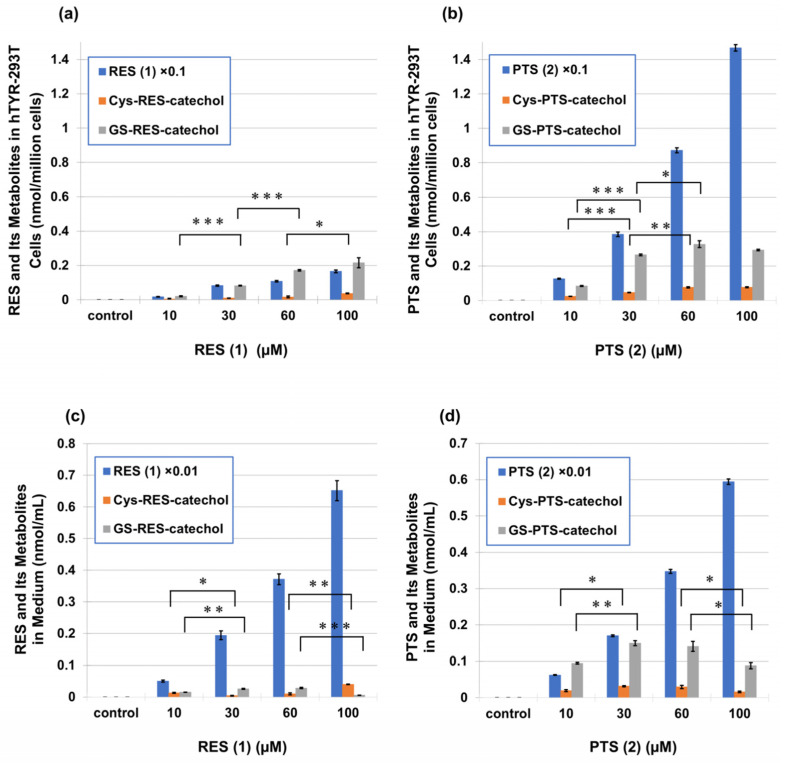
Metabolism of RES (**1**) and PTS (**2**) in hTYR-293T cells yielding their adducts with CySH and GSH. (**a**) RES (**1**) and its metabolites in cells. (**b**) PTS (**2**) and its metabolites in cells. (**c**) RES (**1**) and its metabolites in the medium. (**d**) PTS (**2**) and its metabolites in the medium. Data represent means ± SD (*n* = 3 wells). Statistically significant differences: * *p* < 0.05, ** *p* < 0.01, *** *p* < 0.001 between each treatment concentration (in µM) at RES (**1**) and PTS (**2**). The statistically significance of the differences was determined by Student’s *t*-test (two-tailed).

**Figure 8 ijms-25-09990-f008:**
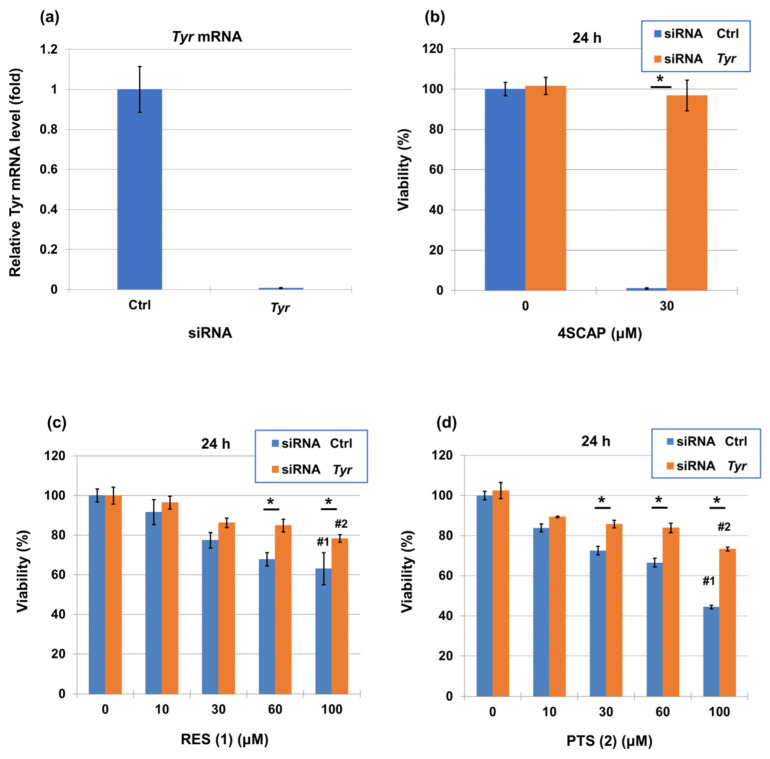
(**a**) Level of *Tyr* mRNA in siRNA-transfected cells. (**b**) 4SCAP, (**c**) RES (**1**), and (**d**) PTS (**2**) induce tyrosinase-dependent reductions in the viability of B16BL6 melanoma cells. B16BL6 melanoma cells were transfected with a negative control siRNA or with a siRNA directed against *Tyr* for 24 h. The cells were then treated with the indicated concentrations of compounds for 24 h, and their viability and mRNA levels (in vehicle-treated cells) were measured. Data represent means ± SD (*n* = 3 wells). The experiments were repeated once, and good reproducibility was obtained. The figure was from a single experiment and representative. * *p* < 0.05 between siRNA control and siRNA *Tyr* at each treatment concentration (in µM). #1: *p* = 0.017 between RES (**1**) and PTS (**2**) in siRNA Ctrl. #2: *p* = 0.020 between RES (**1**) and PTS (**2**) in siRNA *Tyr* at 100 µM, respectively.

**Figure 9 ijms-25-09990-f009:**
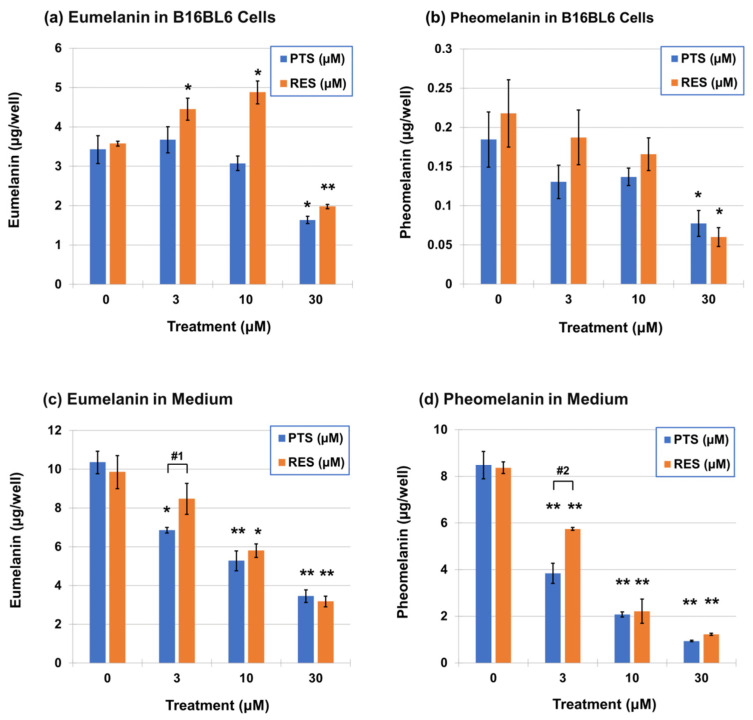
(**a**,**b**): Eumelanin and pheomelanin contents in B16BL6 melanoma cells. Data represent means ± SD (*n* = 3 wells). *p* value between control (0 μM) and each treatment concentration (µM) at eumelanin and pheomelanin in B16BL6 melanoma cells. Statistically significant differences: * *p* < 0.01, ** *p* < 0.001. (**c**,**d**): Eumelanin and pheomelanin contents in the medium. *p* value between control (0 μM) and each treatment concentration (µM) at eumelanin and pheomelanin in the medium. * *p* < 0.01, ** *p* < 0.001. #1: *p* = 0.026 between RES (**1**) and PTS (**2**) in eumelanin values at 3 µM. #2: *p* = 0.002 between RES (**1**) and PTS (**2**) in pheomelanin values at 3 µM.

## Data Availability

Data are contained within the article and Appendix A.

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
