# Peer review of "Pterostilbene, a Dimethyl Derivative of Resveratrol, Exerts Cytotoxic Effects on Melanin-Producing Cells through Metabolic Activation by Tyrosinase"

_ijms, 2024, doi:10.3390/ijms25189990_

Round 1

Reviewer 1 Report

Comments and Suggestions for Authors

Hitomi Tanaka and coworkers report that the polyphenol pterostilbene, which is present in large amounts in blueberries, has a cytotoxic effects on cells expressing the pigment-forming enzyme tyrosinase. Pterostilbene was more cytotoxic than the structurally similar resveratrol, and it blocked tyrosinase-dependent pigment formation. This is a very interesting manuscript. The study was well designed. The presentation of the data is good. The discussion of results is fine.  

Comments:

1. The title, abstract and text refer to “melanocytes”, but actually experiments were not performed in normal melanocytes. Is it correct to claim effects on melanocytes in general if an immortalized melanocyte line was studied? If the authors want to conclude on melanocytes, primary melanocytes should be investigated. The comparison of normal melanocytes versus melanoma cells would be interesting. However, this comparison is not necessary. However, the wording should be correct.

2. The legends of Figures 3, 4 and 4 contain the statement, “reproducibility was confirmed for each experiment”. Please provide more information about the number of experiments. Are the data in the figure representative for all experiments?

3. Mushroom tyrosinase was used for experiments in vitro. Please explain whether the tyrosinase from mushroom has the same properties as mammalian tyrosinase. If not, what might be the impact of differences?

4. In the Discussion, different melanoma cell lines with similar names are mentioned: B16BL6, B16F1, B16F10, B164A5. To avoid confusion, can you explain the meaning of these names, or indicate if they should be considered the same?

Author Response

We appreciate two reviewers for their valuable comments on our manuscript. We revised the manuscript according to their comments as follows.

Reviewer 1

Comments and Suggestions for Authors

Hitomi Tanaka and coworkers report that the polyphenol pterostilbene, which is present in large amounts in blueberries, has a cytotoxic effects on cells expressing the pigment-forming enzyme tyrosinase. Pterostilbene was more cytotoxic than the structurally similar resveratrol, and it blocked tyrosinase-dependent pigment formation. This is a very interesting manuscript. The study was well designed. The presentation of the data is good. The discussion of results is fine.  

Comments:

  1. The title, abstract and text refer to “melanocytes”, but actually experiments were not performed in normal melanocytes. Is it correct to claim effects on melanocytes in general if an immortalized melanocyte line was studied? If the authors want to conclude on melanocytes, primary melanocytes should be investigated. The comparison of normal melanocytes versus melanoma cells would be interesting. However, this comparison is not necessary. However, the wording should be correct.

We agree to the reviewer’s comment. The title has been changed to “Pterostilbene, a dimethyl derivative of resveratrol, exerts cytotoxic effects on melanin-producing cells through metabolic activation by tyrosinase.“

The production of highly reactive o-quinone metabolites by tyrosinase is expected to cause melanocyte-specific damage. For the assessment of tyrosinase-dependent cytotoxicity, we used melanoma B16BL6 cells, but not human melanocytes, because it seemed difficult to consistently obtain human melanocytes with the same properties.

Rhododendrol (RD), one of leukoderma-inducing 4-substituted phenols, is metabolized by tyrosinase, activated to an o-quinone metabolite, and induces tyrosinase-dependent cytotoxicity both in human melanocytes [1,2] and in melanoma cells [2,3]. However, melanocytes derived from different doners exhibit different response to RD [1]. There were huge differences in RD concentrations causing cytotoxicity in different lots of melanocytes: four melanocytes exhibited cytotoxicity at micromolar range of RD, whereas millimolar range of RD was required to induce cytotoxicity in nine melanocytes [1]. We therefore used B16BL6 melanoma cells as a model cell system. We found that tyrosinase expression was efficiently diminished by transfection of a specific siRNA in this cell line, and that some leukoderma-inducing phenols exhibited tyrosinase-dependent cytotoxicity [4].

Reference

1) Kasamatsu S, Hachiya A, Nakamura S, Yasuda Y, Fujimori T, Takano K, et al. Depigmentation caused by application of the active brightening material, rhododendrol, is related to tyrosinase activity at a certain threshold. J Dermatol Sci. 2014; 76: 16–24.

2) Sasaki M, Kondo M, Sato K, Umeda M, Kawabata K, Takahashi Y, et al. Rhododendrol, a depigmentation-inducing phenolic compound, exerts melanocyte cytotoxicity via a tyrosinase-dependent mechanism. Pigment Cell Melanoma Res. 2014; 27: 754–63.

3) Ito S, Okura M, Nakanishi Y, Ojika M, Wakamatsu K, Yamashita T. Tyrosinase-catalyzed metabolism of rhododendrol (RD) in B16 melanoma cells: production of RD-pheomelanin and covalent binding with thiol proteins. Pigment Cell Melanoma Res. 2015; 28: 295–306.

4) Nishimaki-Mogami, T.; Ito, S.; Cui, H.; Akiyama, T.; Tamehiro, N.; Adachi, R.; Wakamatsu, K.; Ikarashi, Y.; Kondo, K. A cell-based evaluation of human tyrosinase-mediated metabolic activation of leukoderma-inducing phenolic compounds. J. Dermatol. Sci. 2022, 108, 77-86.

  1. The legends of Figures 3, 4 and 5 contain the statement, “reproducibility was confirmed for each experiment”. Please provide more information about the number of experiments. Are the data in the figure representative for all experiments?

I added the below sentences in Figure legend in Figures 3, 4, 5, and 6.

“The experiments were repeated once, and good reproducibility was obtained. The figure was from a single experiment and representative.”

  1. Mushroom tyrosinase was used for experiments in vitro. Please explain whether the tyrosinase from mushroom has the same properties as mammalian tyrosinase. If not, what might be the impact of differences?

Tyrosinase catalyzes hydroxylation of p-phenols to form o-diphenols and oxidation of o-diphenols to o-quinones. This applies not only to mushroom tyrosinase but also to mammalian tyrosinases.

  1. In the Discussion, different melanoma cell lines with similar names are mentioned: B16BL6, B16F1, B16F10, B164A5. To avoid confusion, can you explain the meaning of these names, or indicate if they should be considered the same?

B16BL6, B16F10, B16F1, and B164A5 are all melanoma cell lines derived from mice, but each has different characteristics and uses. B16BL6 originated as a cell line derived from a melanoma in a C57BL/6 mouse. Its malignancy is high, and it has a particularly strong metastatic ability. Its characteristics include the highest malignancy and a strong tendency to metastasize, especially to the lungs, making it a very important model for studying cancer metastasis. B16F10 originates from melanoma cells derived from C57BL/6 mice. Its malignancy is the second highest after B16BL6, and it also has a high metastatic potential. Its characteristics make it a model for evaluating the effects of immunotherapy and new anticancer drugs, and it is particularly useful for metastasis research. B16F1 originates from a C57BL/6 mouse, but is a cell line derived from a primary tumor. Its malignancy is relatively low among the B16 lineage, and it has low metastatic potential. Its characteristics make it suitable for basic cancer research and early evaluation of treatments, and it is also suitable for immunotherapy research. B164A5 originates from a part of the B16 cell lineage, and is used for specific research purposes. Its malignancy is low, like B16F1, but it is often used to study specific cell biology properties. Its characteristics make it a model for evaluating the effects of new drugs in some studies. For research purposes, B16BL6 is highly malignant, making it useful for cancer progression and metastasis research, as well as for new drug development. B16F10 is particularly useful for evaluating immunotherapy and anticancer drugs, taking advantage of its high metastatic potential. B16F1 is used for skeletal cancer research and early treatment exploration. B164A5 is often used as an option depending on special research needs. In terms of biological properties, B16BL6 is characterized by high metastatic and malignant potential. B16F10 has high metastatic potential, but is slightly less malignant than B16BL6. B16F1 has more stable properties and is ideal for basic research and early evaluation. B164A5 is suitable for exploring specific cell biology properties and treatments. Generally, these cell lines are well-known and commonly used in research, so most papers do not bother to describe these cell lines. Researchers use the cells that are most appropriate for their research. Therefore, I do not think it is necessary to explain these cell lines one by one in this paper.

We would appreciate your understanding and approval of the above points.

Minor comments:

1) Space should go after “(1)”, similar double space / no space issues should be corrected in the manuscript.

Many thanks. I corrected them.

2) Figures 3,4 and 5 – subfigures A and B are of a different size.

Many thanks. I corrected these figures.

3) Figures 3,4 and 5 – absorbance values are exceeding 1 thus the lines observed might be distorted due to the limitations of Lambert-Beer’s law.

Many thanks for your suggestion. As you mentioned, the relationship between concentration and absorbance is known to be most linear between absorbance 0 and 1. When the absorbance was measured at PTS concentrations of 100 µM, 75 µM, 50 µM, 25 µM, and 0 µM at a maximum wavelength of 310 nm, it was 2.13, 1.77, 1.37, and 0.79, respectively. Indeed, a good linear relationship was observed between 0 and 25 µM. However, this absorption spectrum was not used to quantify the concentration in this study, but rather to track the generation of quinone chromophore. Compared to the absorbance at 310 nm, the absorbance of the quinoid chromophore at 450-500 nm is about 0.5 to 0.2. If the maximum absorbance at 310 nm is set to be less than 1, it was difficult to check the absorption of the quinoid chromophore. Thus, we measured the time course of the tyrosinase-catalyzed oxidation of 100 µM PTS.

The following paper also reports spectra with maximum absorbance values of 3–4 to measure the formation of quinone chromophores:

Reference

1) Xiao-Xin, Chen et al. Condensed Tannins from Ficus virens as Tyrosinase Inhibitors: Structure, Inhibitory Activity and Molecular Mechanism. Plos One, 2014, 9, e91809.

2) Doyle G. Graham, Peter W. Jeffs. The Role of 2,4,5-Trihydroxyphenylalanine in Melanin Biosynthesis. J. Biol. Chem., 1977, 252, 5729-5734.

3) Haibo, Yang et al. Anti-tyrosinase and antioxidant activity of proanthocyanidins from Cinnamomum camphora. Int. J. Food Prop., 2012, 24, 1265-1278.

4) Figure 7 – please explain what “a” and “b” on the graphs stand for. Why Student’s t-test and not ANOVA was used for statistical analysis?

Many thanks for your suggestion. I changed the sentence in legend of Figure 7 as follows:

Metabolism of RES (1) and PTS (2) in hTYR-293T cells yielding their adducts with CySH and GSH. (a) RES (1) and its metabolites in cells. (b) PTS (2) and its metabolites in cells. (c) RES (1) and its metabolites in the medium. (d) PTS (2) and its metabolites in the medium. Data represent means ± SD (n = 3 wells). Statistically significant differences: *p < 0.05, **p < 0.01, ***p < 0.001 between each treatment concentration (in µM) at RES (1) and PTS (2). The statistically significance of the differences was determined by Student’s t-test (two-tailed).

ANOVA is a test to examine whether there is a statistical difference between the means of three or more independent groups. In this case, we wanted to examine whether there was a difference between the two samples, so we used t-teat.

5) Figure 8 – subfigures are not aligned, please correct the issue.

Many thanks. I corrected the subfigures.

6) Some crossed letters can be found through the text, please correct.

Many thanks. I corrected them.

Reviewer 2 Report

Comments and Suggestions for Authors

The work by Tanaka et al focuses on the cytotoxic effect of pterostilbene on melanoma cell line B16BL6. Although the work is quite interesting, suggesting a high potential of pterostilbene against that melanoma cell line, some adjustments have to be made before the work could be approved for IJMS.

General comments:

1)      Using the phrase “melanocytes” in case of B16BL6 cells is not right, as it suggest that normal melanocytes are used and not cancerous cells. Also using that phrase in the title seems to weaken the outcome of the research. Please correct the title, and where applicable, the manuscript.

2)      Figures 3,4 and 5 – absorbance values are exceeding 1 thus the lines observed might be distorted due to the limitations of Lambert-Beer’s law. Please explain.

3)      What was the number of biological replicates used for each experiment? Were there 3 independent experiments performed with 3 triplicates?

4)      Can the concentrations used in biological studies, especially above 30 uM, be applied against melanoma cells in real-life conditions?

5)      Was the overall level of melanins determined for examined samples?

Minor comments:

1)      Space should go after “(1)”, similar double space / no space issues should be corrected in the manuscript

2)      Figures 3,4 and 5 – subfigures A and B are of a different size

3)      Figures 3,4 and 5 – absorbance values are exceeding 1 thus the lines observed might be distorted due to the limitations of Lambert-Beer’s law.

4)      Figure 7 – please explain what “a” and “b” on the graphs stand for. Why Student’s t-test and not ANOVA was used for statistical analysis?

5)      Figure 8 – subfigures are not aligned, please correct the issue

6)      Some crossed letters can be found through the text, please correct.

Author Response

We appreciate two reviewers for their valuable comments on our manuscript. We revised the manuscript according to their comments as follows.

Reviewer 2:

Comments and Suggestions for Authors

The work by Tanaka et al focuses on the cytotoxic effect of pterostilbene on melanoma cell line B16BL6. Although the work is quite interesting, suggesting a high potential of pterostilbene against that melanoma cell line, some adjustments have to be made before the work could be approved for IJMS.

General comments:

  • Using the phrase “melanocytes” in case of B16BL6 cells is not right, as it suggest that normal melanocytes are used and not cancerous cells. Also using that phrase in the title seems to weaken the outcome of the research. Please correct the title, and where applicable, the manuscript.

We agree to the reviewer’s comment. The title has been changed to “Pterostilbene, a dimethyl derivative of resveratrol, exerts cytotoxic effects on melanin-producing cells through metabolic activation by tyrosinase.“

The production of highly reactive o-quinone metabolites by tyrosinase is expected to cause melanocyte-specific damage. For the assessment of tyrosinase-dependent cytotoxicity, we used melanoma B16BL6 cells, but not human melanocytes, because it seemed difficult to consistently obtain human melanocytes with the same properties.

Rhododendrol (RD), one of leukoderma-inducing 4-substituted phenols, is metabolized by tyrosinase, activated to an o-quinone metabolite, and induces tyrosinase-dependent cytotoxicity both in human melanocytes [1,2] and in melanoma cells [2,3]. However, melanocytes derived from different doners exhibit different response to RD [1]. There were huge differences in RD concentrations causing cytotoxicity in different lots of melanocytes: four melanocytes exhibited cytotoxicity at micromolar range of RD, whereas millimolar range of RD was required to induce cytotoxicity in nine melanocytes [1]. We therefore used B16BL6 melanoma cells as a model cell system. We found that tyrosinase expression was efficiently diminished by transfection of a specific siRNA in this cell line, and that some leukoderma-inducing phenols exhibited tyrosinase-dependent cytotoxicity [4].

Referecnces:

1) Kasamatsu S, Hachiya A, Nakamura S, Yasuda Y, Fujimori T, Takano K, et al. Depigmentation caused by application of the active brightening material, rhododendrol, is related to tyrosinase activity at a certain threshold. J Dermatol Sci. 2014; 76: 16–24.

2) Sasaki M, Kondo M, Sato K, Umeda M, Kawabata K, Takahashi Y, et al. Rhododendrol, a depigmentation-inducing phenolic compound, exerts melanocyte cytotoxicity via a tyrosinase-dependent mechanism. Pigment Cell Melanoma Res. 2014; 27: 754–63.

3) Ito S, Okura M, Nakanishi Y, Ojika M, Wakamatsu K, Yamashita T. Tyrosinase-catalyzed metabolism of rhododendrol (RD) in B16 melanoma cells: production of RD-pheomelanin and covalent binding with thiol proteins. Pigment Cell Melanoma Res. 2015; 28: 295–306.

4) Nishimaki-Mogami, T.; Ito, S.; Cui, H.; Akiyama, T.; Tamehiro, N.; Adachi, R.; Wakamatsu, K.; Ikarashi, Y.; Kondo, K. A cell-based evaluation of human tyrosinase-mediated metabolic activation of leukoderma-inducing phenolic compounds. J. Dermatol. Sci. 2022, 108, 77-86.

2) Figures 3,4 and 5 – absorbance values are exceeding 1 thus the lines observed might be distorted due to the limitations of Lambert-Beer’s law. Please explain.

Many thanks for your suggestion. As you mentioned, the relationship between concentration and absorbance is known to be most linear between absorbance 0 and 1. When the absorbance was measured at PTS concentrations of 100 µM, 75 µM, 50 µM, 25 µM, and 0 µM at a maximum wavelength of 310 nm, it was 2.13, 1.77, 1.37, and 0.79, respectively. Indeed, a good linear relationship was observed between 0 and 25 µM. However, this absorption spectrum was not used to quantify the concentration in this study, but rather to track the generation of quinone chromophore. Compared to the absorbance at 310 nm, the absorbance of the quinoid chromophore at 450-500 nm is about 0.5 to 0.2. If the maximum absorbance at 310 nm is set to be less than 1, it was difficult to check the absorption of the quinoid chromophore. Thus, we measured the time course of the tyrosinase-catalyzed oxidation of 100 µM PTS.

The following paper also reports spectra with maximum absorbance values of 3–4 to measure the formation of quinone chromophores:

1) Xiao-Xin, Chen et al. Condensed Tannins from Ficus virens as Tyrosinase Inhibitors: Structure, Inhibitory Activity and Molecular Mechanism. Plos One, 2014, 9, e91809.

2) Doyle G. Graham, Peter W. Jeffs. The Role of 2,4,5-Trihydroxyphenylalanine in Melanin Biosynthesis. J. Biol. Chem., 1977, 252, 5729-5734.

3) Haibo, Yang et al. Anti-tyrosinase and antioxidant activity of proanthocyanidins from Cinnamomum camphora. Int. J. Food Prop., 2012, 24, 1265-1278.

3) What was the number of biological replicates used for each experiment? Were there 3 independent experiments performed with 3 triplicates?

Regarding Figure 7, 8, and S16, (transfection) experiments were performed in three separate wells (n=3), and regarding Figure 8, similar results were obtained in two independent experiments. Regarding Figure S16, similar results were obtained in two or three experiments.

Regarding Figure 9, cells from three wells were analyzed as biological replicates (n = 3) for each treatment.

4) Can the concentrations used in biological studies, especially above 30 uM, be applied against melanoma cells in real-life conditions?

PTS and RES (60 mM and 100 mM) can be applied to melanoma cells. The stock solutions in DMSO (200x) were diluted into the medium containing 10% FBS. Final concentration of DMSO was 0.5%.

This condition was described in the original manuscript [section 4.7]

5) Was the overall level of melanins determined for examined samples?

It has been reported that eumelanin and pheomelanin can be obtained by multiplying PTCA and 4-AHP by 25 and 9 times, respectively, so we created new graphs of these data [1,2,3].

Reference

  1. Ito, S.; Nakanishi, Y.; Valenzuela, R.K.; Brilliant, M.H.; Kolbe, L.; Wakamatsu, K. Usefulness of alkaline hydrogen peroxide oxidation to analyze eumelanin and pheomelanin in various tissue samples: Application to chemical analysis of human hair melanins. Pigment Cell Melanoma Res. 2011, 24, 605–613. doi: 10.1111/j.1755-148X.2011.00864.x.
  2. Wakamatsu, K.; Ito, S.; Rees, J.L. The usefulness of 4-amino-3-hydroxyphenylalanine as a specific marker of pheomelanin. Pigment Cell Melanoma Res. 2002, 15, 225–232. doi: 10.1034/j.1600-0749.2002.02009.x.
  3. d’Ischia, M.; Wakamatsu, K.; Napolitano, A.; Briganti, S.; Garcia-Borron, J.-C.; Kovacs, D.; Meredith, P.; Pezzela, A.; Picardo, M.; Sarna, T.; et al. Melanins and melanogenesis: methods, standards, protocols. Pigment Cell Melanoma Res. 2013, 26, 616-633. doi: 10.1111/pcmr.12121.

Round 2

Reviewer 2 Report

Comments and Suggestions for Authors

I want to thank the authors for responding to all of the addressed issues. I believe that the article should be accepted in in its recent forms. During the proofing, please mind that Figure 9 was probably not replaced by the new one, but is overlayed on the old one, as can be deduced from what I can see on the left side of the graph.